# `BSemiFL`: Semi-supervised Federated Learning via a Bayesian Approach

**Haozhao Wang** [1]   **Shengyu Wang** [2]   **Jiaming Li** [1]   **Hao Ren** [3]   **Xingshuo Han** [4]   **Wenchao Xu** [2]   **Shangwei Guo** [5]
**Tianwei Zhang** [6]   **Ruixuan Li** [1]

## Abstract

Semi-supervised Federated Learning (SSFL) allows clients to collaboratively train a global model in the absence of their local data labels. The key step of SSFL is the re-labeling where each client adopts two types of available models, namely global and local models, to re-label the local data. While various technologies such as using the global model or the average of two models have been proposed to conduct the re-labeling step, little literature delves deeply into the performance dominance and limitations of the two models. This paper first *theoretically* and *empirically* demonstrate that the local model achieves higher re-labeling accuracy over local data while the global model can progressively improve the re-labeling performance by introducing the extra knowledge of other clients. Based on these, we propose `BSemiFL` which re-labels the local data via the collaboration between the local and global model in a Bayesian approach. Specifically, to re-label any given local sample, `BSemiFL` first uses Bayesian inference to assess the closeness of the local/global model to the sample. Then, it applies a weighted combination of their pseudo labels, using the closeness as the weights. Theoretical analysis shows that the labeling error of our method is smaller than that of simply using the global model, the local model, or their simple average. Experiments show that `BSemiFL` improves the performance by up to $9.8\%$ as compared to existing methods.

---

[1]School of Computer Science and Technology, Huazhong University of Science and Technology, Wuhan, China [2]Division of Integrative Systems and Design, Hong Kong University of Science and Technology, Hongkong, China [3]School of Cyber Science and Engineering, Sichuan University, Chengdu, China [4]College of Computer Science and Technology, Nanjing University of Aeronautics and Astronautics, Nanjing, China [5]College of Computer Science, Chongqing University, Chongqing, China [6]College of Computing and Data Science, Nanyang Technological University, Singapore, Singapore. Correspondence to: Ruixuan Li <rxli@hust.edu.cn>.

*Proceedings of the 42$^{nd}$ International Conference on Machine Learning*, Vancouver, Canada. PMLR 267, 2025. Copyright 2025 by the author(s).

## 1. Introduction

Federated learning (FL) enables the collaboration of multiple clients to train a global model without sharing their private dataset (Hu et al., 2024a; Huang et al., 2024a; Qi & et al., 2025; Liu et al., 2024), gaining much attention recently and being applied to a wide range of applications such as traffic prediction(Zhang et al., 2024a), recommendation (Yang et al., 2024), and IoT (Liu et al., 2021a). However, typical FL (McMahan et al., 2017) presumes that each client possesses labels for their data, a requirement which may not align with practical realities in numerous applications, especially when most clients may not be experts in the task of interest or may not be well motivated to label their data. To tackle this challenge, Semi-supervised FL (SSFL) (Diao et al., 2022; Jeong et al., 2021; Zhuang et al., 2021) comes to the rescue, *by allowing clients to have access to purely unlabeled data while maintaining a small amount of labeled data in the server*.

SSFL typically runs the re-labeling and local training steps in each client and then conducts the aggregation step in the server, as shown in Figure 1. Although the process is easy to implement, the re-labeling error of the local data is inevitable, thus degrading the performance. To improve the SSFL performance, many approaches have been proposed from the re-labeling step (Diao et al., 2022), local training step (Malaviya et al., 2023; Lee et al., 2024), and aggregation step (Wang et al., 2023b) separately. In this paper, we mainly focus on the re-labeling step by considering that it can directly reduce the error of pseudo labels.

The representative re-labeling technique (Diao et al., 2022) is to adopt the global model to re-label the local data. While easy to implement, the upper bound of its performance is inevitably limited by adaptation error of global model, which arises from the substantial gap between the global and the local data distribution (Hu et al., 2024b; Huang et al., 2024b; Qi et al., 2023; Wang et al., 2023a; Tang et al., 2022). To reduce the adaptation error, a natural solution is to adopt the local model re-label the local data, where the local model is trained with the local data and adapts better than the global model. Yet, we find that merely utilizing the local model leads to a less optimal global model than using the global model. We identify that the underlying principle behind this is that the data knowledge embodied by the local model is inherently solidified to its own small

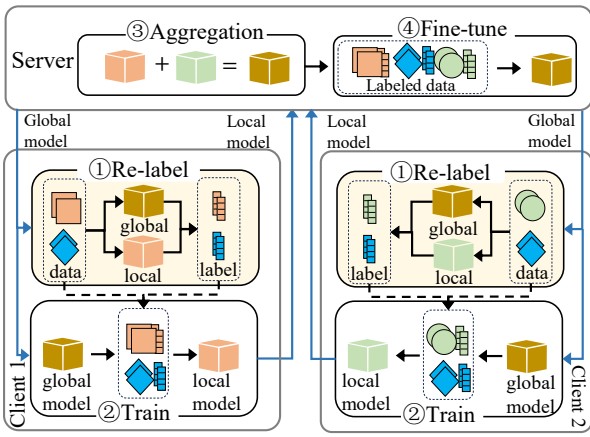

*Figure 1.* The framework of SSFL.

local dataset, which cannot be optimized by referring extra knowledge from other clients.

Considering the above limitations, we propose BSemiFL which uses a Bayesian-based approach to jointly utilize the ensemble of the global and local model to label the local data to harness both their benefits. Instead of adopting a naively averaging ensemble where the correctness of pseudo labels heavily depends on the prediction of both models, BSemiFL performs the ensemble in a weighted manner where the weights are adaptively calculated for each sample using a Bayesian approach. More specifically, BSemiFL adopts the Bayesian approach to infer the closeness of the local sample to the training data distribution of the local or global model and sets the weights based on the closeness. The principle behind this is that the model tends to correctly predict the label of a sample when the given sample is closer to its training data. Therefore, BSemiFL can assign a higher weight to the global/local model when the given sample is closer to the global/local training data, thus being able to reduce the impact of wrong pseudo labels. Extensive experiments show that the BSemiFL significantly improves the model accuracy as compared to state-of-the-art algorithms. Our contributions are:

- As far as we know, this is *the first work that theoretically reveals why merely adopting the global or local model is not optimal for labeling local data*, which are also validated by our empirical evaluations. Specifically, the main causes are that the distribution gap between the global data and local data limits the performance of the global model over the local data and the error of the local model can not be corrected by is local data.

- We propose a novel method called BSemiFL which jointly leverages the global and local model to label the local data via a Bayesian approach. In particular, we employ a Bayesian-based approach to adaptively discern

the closeness of the global and local model to each local sample, and then we apply a weighted ensemble of the global and local model for labeling the local data where the weights are the obtained confidence.

- We establish the theories for re-labeling performance of the proposed method. Our results show that BSemiFL theoretically reduces the labeling error as compared to the maximum error of merely adopting either the global or local model. Besides, we also theoretically show that our Bayesian-based weighted ensemble achieves a smaller loss than naively averaging.

- We conduct extensive experiments across a diverse array of datasets and experimental configurations. The outcome robustly substantiates the superiority of our proposed method, demonstrating an improvement in model accuracy that reaches up to $9.8\%$ compared with the current state-of-the-art methodologies.

## 2. Related Work

### 2.1. Semi-supervised Federated Learning

Existing SSFL frameworks can be categorized into two types according to the location of the labeled data, i.e., "*labels at client*" (Zhang et al., 2024b; Bai et al., 2024; Zhu et al., 2024) and "*labels at server*". SSFL of "labels at client" considers labeled data are available at local clients (Kim et al., 2023; Itahara et al., 2023; Xu et al., 2024; Zhang et al., 2023a;b; Cho et al., 2023). The first work of this type FedMatch (Jeong et al., 2021) assumes that each client contains both labeled and unlabeled data and employs a consistency method from semi-supervised learning, enhancing cross-model intra-client consistency by selecting proxy helpers among the clients. FedAC (Jiang et al., 2024) identifies that using the models of similar and dissimilar clients to label the unlabeled data has different advantages, thus proposing alternatively leveraging the models of similar and dissimilar clients to label the unlabeled data, which demonstrates great effectiveness. Considering the discrepancy of clients that some clients may not be able to label the data while other clients are available, some works (Lin et al., 2021; Liang et al., 2022; Liu et al., 2021b) also focus on the setting where clients are fully labeled while some clients only contain unlabeled samples.

In this paper, *we focus on the second type of SSFL, i.e., "labels at server",* by considering that it is generally practical in real-world FL scenarios, e.g., in the healthcare field where the labels are mostly assigned by experts and users usually are not able to label their health monitoring data (Lee et al., 2024; Malaviya et al., 2023; Yang Xu, 2023; Yang et al., 2023). SSFL of "labels at server" assumes that clients have purely unlabeled data with a limited amount of labels residing at the central server ((Lin et al.,

2021; He et al., 2021; Zhang et al., 2021)); The representative work SemiFL (Diao et al., 2022) innovates with an alternative training regimen that re-trains the global model in the server and labels the local data with the global model. Being different from SemiFL, we leverage an Bayesian-based collaboration between the global model and the local model to re-label local data. A subsequent work (FL)$^2$ (Lee et al., 2024) utilizes the idea of contrastive learning to improve the local training process. Considering that the client may be resource-constrained, pFedKnow (Wang et al., 2023b) proposes compressing the local models to reduce the computation and communication cost of SSFL. Different from these works, this paper focuses on the re-labeling step, which is orthogonal to them.

## 2.2. Unsupervised Federated Learning

Unsupervised federated learning (UFL) does not assume the existence of labeled data in either the server or clients during the training process. Since contrastive learning is the main technology for learning in unlabeled data (Li et al., 2021; Zheng et al., 2022), many works (Zhuang et al., 2021; 2022) propose applying the contrastive learning methods over the scenario of UFL. For example, FedU (Zhuang et al., 2021) allows each client to adopt a conservative learning loss during the local training process. Orchestra (Lubana et al., 2022) addresses the challenge of NonIID data by adopting a client clustering approach, thereby mitigating the adverse effects caused by data heterogeneity among clients. These works typically focus on Step 2 as illustrated in Figure 1, which is orthogonal to this work that seeks to improve Step 1.

## 3. Problem Formulation and Preliminaries

**Problem Formulation**. Semi-supervised federated learning is to collectively train a global model $\mathbf{w}$ that adapts to the global data distributed in $M$ clients. In SSFL, the server has access to a labeled dataset $\mathbb{S} = \{(x_i, y_i)\}_{i=1}^{N_s}$ with $N_s$ samples. Each client $m$ has access to a distinct unlabeled dataset $\mathbb{U}_m = \{x_i^m\}_{i=1}^{N_m}$ with $N_m$ samples where the true label $y_i^m$ does not exist. We consider a $K$-class classification task in the SSFL and denote $f(x, \mathbf{w})$ by the $K$-dimensional soft prediction output by the model $\mathbf{w}$ using the softmax function.

**Basic workflow of SSFL**. For simplicity and clarity, we here specify the basic four steps of SSFL, as illustrated in Figure 1. In each round $t$, after receiving the global model $\mathbf{w}_t$ from the server, corresponding to ① in Figure 1, each selected client $m$ starts to generate pseudo labels $\hat{y}_i^m$ for each local sample $x_i$ by using the model $\tilde{\mathbf{w}}_t^m$:

$$\hat{y}_i^m = f(x_i^m, \tilde{\mathbf{w}}_t^m), \quad (1)$$

where the adopted re-labeling model can either be the

global model $\mathbf{w}_t$, local model $\mathbf{w}_t^m$, or be their combinations. To construct a high-confidence dataset $\mathbb{S}_m$, each client uses a threshold to filter out labels with low confidence values like FixMatch (Sohn et al., 2020):

$$\mathbb{S}_m = \{(x_i^m, \mathbb{I}(\hat{y}_i^m)) \quad \text{with} \quad \max(\hat{y}_i^m) \geq \tau\}, \quad (2)$$

where $\mathbb{I}(\cdot)$ is to transform the vector $\hat{y}_i^m$ into the one-hot label vector with only remaining the maximum probability. $0 < \tau < 1$ is a threshold pre-selected by all clients. $\max(\hat{y}_i^m)$ denotes the maximum confidence among all $K$ classes. With the constructed labeled dataset, as illustrated in ② of Figure 1, client $m$ trains its local model for $E$ local epochs to obtain the local model:

$$L_m = l(f(x_i^m, \mathbf{w}_t), \mathbb{I}(\hat{y}_i^m)), \quad \mathbf{w}_t^m = \mathbf{w}_t^m - \eta \nabla_{\mathbf{w}_t^m} L_m. \quad (3)$$

After that, as presented by ③ in Figure 1, each client $m$ uploads its local model $\mathbf{w}_t^m$ to the server and server aggregates them to obtain the $t+1$-th round global model

$$\mathbf{w}_{t+1} = \sum_{m=1}^{M_t} \frac{S_m}{S^t} \mathbf{w}_t^m, \quad (4)$$

where $S^t = \sum_{m=1}^{M_t} S_m$ and $S_m$ denotes the size of $\mathbb{S}_m$. Finnally, corresponding to ④ in Figure 1, the server fine-tunes the global model $\mathbf{w}_{t+1}$ with the loss $L_s$ for $E$ epochs:

$$L_s = l(f(x_i, \mathbf{w}_{t+1}), y_i), \mathbf{w}_{t+1} = \mathbf{w}_{t+1} - \eta \nabla_{\mathbf{w}_{t+1}} L_s, \quad (5)$$

where $l$ represents the loss function (e.g., Cross Entropy loss) and $\eta$ denotes the learning rate.

## 4. Motivation

This section specifies the limitations and benefits of the global model and local model for re-labeling the local data. For simplicity, we below denote the SSFL method of re-labeling with only the global model by 'global-only' and with only the local model by 'local-only'. In the following, we conduct motivation experiments by using the Wide ResNet28 × 2 model and CIFAR-10 dataset.

### 4.1. Limitations and benefits of the global model

**Limitations of the global model**. Specifically, *the global model inherently has generalization errors on local data, which stem from the differences between the global data distribution and the local data distribution due to statistical heterogeneity.* To validate this, we run the global-only method over CIFAR-10 dataset to compare the average accuracy of the global model over the local data of all clients to that of the local models over their corresponding local data, where the results are shown in Figure 2a. Noting that the test data for the global model is the global data, while

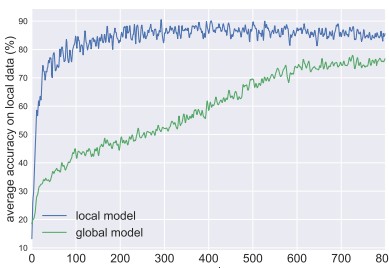 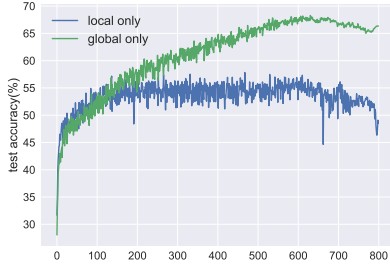 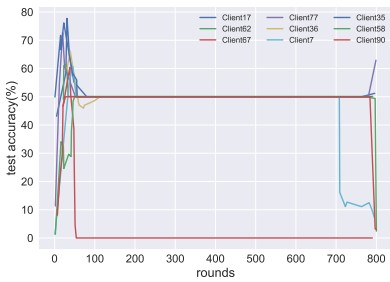

(a) Average local accuracy of the global and local model on local data separately

(b) Global accuracy of the global model obtained by local-only & global-only method

(c) Local accuracy of each local model on its corresponding local data

*Figure 2.* Motivation experiments. (a) The average accuracy of testing the global and local model on local data of all clients separately when using the global-only method. (b) Comparison between the local-only and global-only methods. The accuracy is to test the global model on the global test data. (c) The accuracy of the local model on its corresponding local data when using the local-only method.

the test data for the local model is the local data corresponding to its respective client. As can be seen, the accuracy of the global model over local data is much lower than the local model. Theoretically, we show that a substantial performance gap exists when only using the global model to label local data.

**Theorem 4.1.** *Denote the data distribution of each client $m$ by $p_m$, the empirical distribution of the global dataset by $\hat{p}$, and the real distribution of the global data by $p$. By using $d_{\mathcal{F}\Delta\mathcal{F}}(\cdot, \cdot)$ to measure the distance between two distributions, given constants $0 < \delta \leq 1$ and $\sigma > 0$, with the probability at least $1 - \delta$, the expected labeling error $L_{p_m}(f_{\hat{p}})$ of the global hypothesis $f_{\hat{p}}$ over the local data distribution $p_m$ is bound by:*

$$L_{p_m}(f_{\hat{p}}) \leq L_{\hat{p}}(f_{\hat{p}}) + \sqrt{\frac{log\frac{2}{\delta}}{2S}} + \frac{1}{2}d_{\mathcal{F}\Delta\mathcal{F}}(p_m, p) + \lambda_g, \quad (6)$$

*where $S = \sum_{m=1}^{M} S_m + N_s$ denotes the total data size of the global dataset, and $\lambda_g = L_p(f_*) + L_{p_m}(f_*)$ with $f_* = \arg\min_{f \in \mathcal{F}} L_p(f) + L_{p_m}(f)$.*

The proof is deferred to Appendix A. Theorem 4.1 indicates that the labeling performance of the global model is substantially limited by the gap $d_{\mathcal{F}\Delta\mathcal{F}}(p_m, p) + \lambda_g$ between the global distribution $p$ and local distribution $p_m$. As a consequence, merely adopting the global model cannot fully unleash the performance potential of SSFL.

**Benefits of the global model**. *The global model is trained by aggregating local models from all clients, where the generalization error of re-labeling local data can be progressively reduced with the training process by bringing extra knowledge.* As can be seen from Figure 2a, the re-labeling accuracy of the global model gradually approaches the local model. Further, we compare the test accuracy between the local-only and global-only method over the

global dataset over CIFAR10. As shown in Figure 2b, the test accuracy of the global-only method is nearly 15% higher than the local-only method, where the global model is tested using two different annotation methods, and the test data used is the global data.

### 4.2. Limitations and benefits of the local model

**The limitation of the local model**. Specifically, re-labeling with the local model suffers from the *local knowledge solidification, where in each round, the client trains the local model based only on the pseudo-labeled data from the previous round, leading to the updated model merely inheriting and retaining the previous model's knowledge without correcting erroneous labels.* To validate this, we record the accuracy of the local model of each client on its local data across rounds. We test the local model using its corresponding local data. As shown in Figure 2c, we randomly present the results of 9 clients for clarity of presentation. As can be seen, there is no change in the accuracy after the initial stage, indicating that the knowledge of the local model has been fixed and its error cannot be gradually corrected by itself. Besides, training the local model in the local-only method relies on the limited local data in terms of size, causing a high generalization error. These also verify that why the test accuracy of the global model obtained by the local-only method is nearly 15% less than the global-only method in Figure 2b. A theoretical verification is present below.

**Theorem 4.2.** *Let the data distribution of client $m$ be $p_m$ and its empirical distribution be $\hat{p}_m$. Given constants $0 < \delta \leq 1$ and $\sigma > 0$, with the probability at least $1 - \delta$, the expected labeling error $L_{p_m}(f_{\hat{p}_m})$ of the local hypothesis $f_{\hat{p}_m}$ over the local data distribution $p_m$ is bound by:*

$$L_{p_m}(f_{\hat{p}_m}) \leq L_{\hat{p}_m}(f_{\hat{p}_m}) + \sqrt{\frac{log\frac{2}{\delta}}{2S_m}}, \quad (7)$$

*where $S_m$ denotes the data size of the labeled local dataset.*

Theorem 4.2 can be directly derived from Hoeffding inequality and we ignore its proofs. Comparing Theorem 4.2 to Theorem 4.1, we can find that the generalization error $\sqrt{\frac{\log \frac{2}{\delta}}{2S_m}}$ of local-only metho is larger than $\sqrt{\frac{\log \frac{2}{\delta}}{2S}}$ of the global-only method due to the limited size of the local data.

**Benefits of the local model**. *The local model adapts better to the distribution of the local data than the global model due to the eliminated distribution gap of training data*. This can be observed by comparing Theorem 4.2 to Theorem 4.1, where the error items $\frac{1}{2}d_{\mathcal{F}\Delta\mathcal{F}}(p_m, p) + \lambda_g$ in Theorem 4.1 does not exists in Theorem 4.2. Figure 2a also presents that the accuracy of the local model on its local data converges faster and higher than the global model. Therefore, how to harness both the benefits of the small distribution gap of the local-only method and the small generalization error of the global method is the key.

## 5. Methodology

In this section, we specify the proposed method which jointly leverages the global and local model to label the local data. The framework of our method is illustrated in Figure 1 and Figure 3. Our method only refines the re-labeling process, corresponding to ① in Figure 1, which is orthogonal to most recent advancements. For clarity of expression, we here mainly specify the labeling process in this section. Overall, the labeling process of BSemiFL consists of two stages, i.e., calculating the closeness of the global/local model on each local sample and then using the confidence values to apply a weighted ensemble of the pseudo labels generated by the two models. The algorithm workflow can be found in Appendix B.

### 5.1. Bayesian-based Closeness

Based on the domain adaptation theory in (Ben-David et al., 2010), a model tends to achieve low generalization error when the test data are sampled from the space close to its training domain. Therefore, we adopt the closeness of the sample to its training data to measure the correctness. Specifically, we utilize the sampling probability $\hat{p}(x)$ to represent the closeness of the given sample $x$ to the global distribution $\hat{p}$. Obviously, a larger $\hat{p}(x)$ indicates that there are more similar samples around the area of the sample $x$, demonstrating that $x$ is closer to the global data. To estimate the probability $\hat{p}(x)$, we leverage the Variational Bayesian Inference to build a negated evidence lower bound (ELBO). Without causing confusion, we leverage $f_{\hat{p}}(k|x)$ as the posterior probability of classifying the sample $x$ into the class $k$. Further, we denote $p(x|k)$ by the probability of generating the sample $x$ conditioned on label $k$. Then, the ELBO is defined as:

$$\sum_{i=1}^{N_s} \log \hat{p}(x_i) - \sum_{i=1}^{N_s}\sum_{k=1}^{K} f_{\hat{p}}(k|x_i) \log \frac{\hat{p}(x_i|k)\hat{p}_i(k)}{f_{\hat{p}}(k|x_i)}. \quad (8)$$

By fixing the posterior probability $f_{\hat{p}}(k|x_i)$, and considering facts

$$\hat{p}(x_i) = \sum_{k=1}^{K} \hat{p}(x_i, k), \quad \sum_{i=1}^{N_s} \hat{p}(x_i|k) = 1, \quad (9)$$

minimizing (8) obtains the optimum of $\hat{p}(x_i|k)$ which satisfies:

$$\hat{p}(x_i|k) = \frac{f_{\hat{p}}(k|x_i)}{\sum_{x_i=1}^{N_s} f_{\hat{p}}(k|x_i)}. \quad (10)$$

We parameterize $f_{\hat{p}}(k|x_i)$ by the global model $\mathbf{w}_t$ which is fine-tuned on the global dataset $\mathbb{S}$ with empirical distribution $\hat{p}$, i.e., $f_{\hat{p}}(k|x_i, \mathbf{w}_t)$. Then, when considering any specific local sample $x_i^m$ in the client $m$, the corresponding closeness of the global model $\mathbf{w}_t$ conditioned on the class $k$ is approximately

$$Q_s^k = \sum_{i=1}^{N_s} f_{\hat{p}}(k|x_i, \mathbf{w}_t), \quad \hat{p}(x_i^m|k) = \frac{f_{\hat{p}}(k|x_i^m, \mathbf{w}_t)}{f_{\hat{p}}(k|x_i^m, \mathbf{w}_t) + Q_s^k}. \quad (11)$$

Further, considering the empirical distribution of the labeled dataset $\mathbb{S}$ in the server as $\hat{p}_s$ and approximating the class prior probability of the server dataset to the global dataset, i.e., $\hat{p}(k) \approx \hat{p}_s(k)$, we have

$$\hat{p}(x_i^m) = \sum_{k=1}^{K} \hat{p}(x_i^m|k)\hat{p}_s(k). \quad (12)$$

To this end, we can obtain the closeness $\hat{p}(x_i^m)$ of the global model $\mathbf{w}_t$ on any local sample $x_i$. Similarly, we can also obtain the closeness $\hat{p}_m(x_i)$ of the local model $\mathbf{w}_t^m$ on any local sample $x_i$ as

$$\hat{p}_m(x_i^m) = \sum_{k=1}^{K} \hat{p}_m(x_i^m|k)\hat{p}_m(k)$$
$$= \sum_{k=1}^{K} \frac{f_{\hat{p}_m}(k|x_i^m, \mathbf{w}_t^m)}{f_{\hat{p}_m}(k|x_i^m, \mathbf{w}_t^m) + Q_m^k}\hat{p}_m(k),$$
$$\text{where} \quad Q_m^k = \sum_{i=1}^{S_m} f_{\hat{p}_m}(k|x_i, \mathbf{w}_t^m). \quad (13)$$

### 5.2. Weighted Ensemble of Global and Local Model

After obtaining the closeness $\hat{p}(x_i^m)$ of the global model and $\hat{p}_m(x_i^m)$ of the local model on the local sample $x_i^m$,

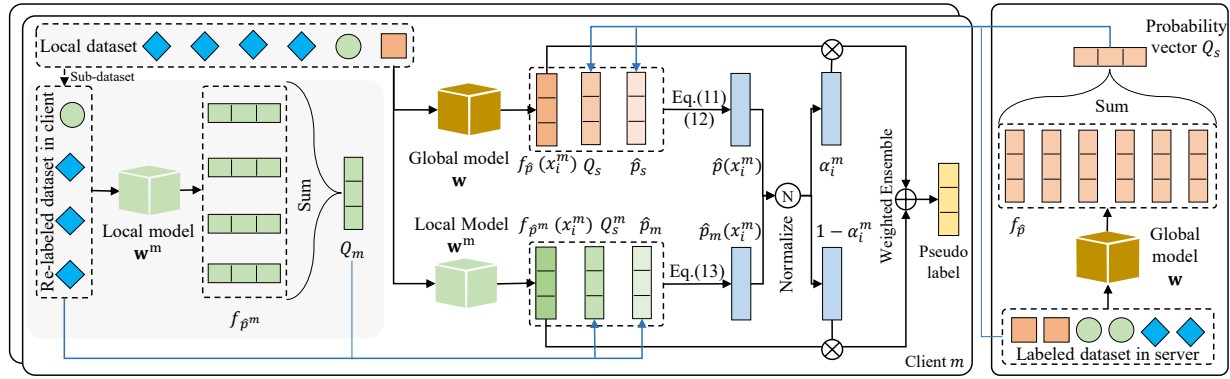

*Figure 3.* Framework of `BSemiFL` with Bayesian-based Ensemble.

the client employs a weighted ensemble of them to generate pseudo labels. Specifically, each client $m$ first normalizes the weights:

$$\alpha_i^m = \frac{\hat{p}(x_i^m)}{\hat{p}(x_i^m) + \hat{p}_m(x_i^m)}, \ 1 - \alpha_i^m = \frac{\hat{p}_m(x_i^m)}{\hat{p}(x_i^m) + \hat{p}_m(x_i^m)}. \tag{14}$$

and then executes:

$$\hat{y}_i^m = \alpha_i^m f_{\hat{p}}(x_i^m, \mathbf{w}_t) + (1 - \alpha_i^m) f_{\hat{p}_m}(x_i^m, \mathbf{w}_t^m). \tag{15}$$

After that, each client $m$ constructs the high-confidence dataset $\mathbb{S}_m$ and performs local training by conducting (2),(3), and (4).

**Privacy Discussion**. Although our method requires more interaction terms, it does not actually increase the risk of privacy leakage. Compared to traditional FL methods, our method only needs to additionally transmit the K-dimensional probability distribution vector $Q$ of the server-side public dataset categories on the global model. In the SSFL scenario, the data on the server side is generally non-private. Therefore, in practice, our method does not result in additional client privacy leakage compared to traditional FL methods.

## 6. Theoretical Analysis

We below theoretically present the benefits of the Bayesian-based ensemble. We initiate by juxtaposing the performance of our method against the global-only or local-only methods. Then, we show how the weighted ensemble surpasses the straightforward averaging method.

**Theorem 6.1.** *Denote the hypothesis trained on the empirical distribution of the local dataset $\hat{p}_m$ by $f_{\hat{p}_m}$ and the hypothesis trained on the empirical distribution of the global dataset $\hat{p}$ by $f_{\hat{p}}$. For any local data sample $x \sim p_m$, considering the Cross-Entropy loss function is adopted, then the loss of the ensemble between the global and local model is*

*less than the maximum individual loss of either the global or local model, i.e.,*

$$L(\alpha f_{\hat{p}} + (1 - \alpha)f_{\hat{p}_m}) \leq \max\left(L(f_{\hat{p}}), L(f_{\hat{p}_m})\right). \tag{16}$$

Proof is in Appendix A. Theorem 6.1 indicates that jointly leveraging the global and local model will inherit the benefits of the better one and thus prevent its performance from becoming the same as the worse one. On the other side, such an approach may restrict itself from outperforming the best model, of which the loss on each sample may be larger than the better one between the global model and the local model. In other words, the prediction probability (confidence value) of the ensemble model on the ground-truth label is smaller than an individual model. However, it is worthwhile to note that a smaller class prediction probability does not mean worse labeling performance. In fact, *we can accordingly adopt a small threshold* (i.e., smaller $\tau$ as specified in equation (2)) when choosing good labels. This has also been verified by our evaluations as shown in Figure 5c, where our optimal threshold is $0.8$ while the optimal threshold of the global-only method is $0.95$ as stated in (Diao et al., 2022). Now, we show that the performance of the weighted ensemble is better than the naive average.

**Theorem 6.2.** *Consider the same notations of distribution and hypothesis defined in previous theorems. For each local sample $x \sim p_m$ with the ground-truth label $y = k$, considering the Cross-Entropy loss function is adopted, then under the condition that $f_{\hat{p}}(k|x) > f_{\hat{p}_m}(k|x)$ when $\hat{p}(x) > \hat{p}_m(x)$ and vice versa, the loss of Bayesian-based model ensemble is smaller than the direct averaging:*

$$L(\alpha f_{\hat{p}} + (1 - \alpha)f_{\hat{p}_m}) \leq L(\frac{f_{\hat{p}} + f_{\hat{p}_m}}{2}). \tag{17}$$

Proof can be found in Appendix A. Theorem 6.2 demonstrates the effectiveness of the proposed Bayesian-based approach. Although this theorem holds under some specific conditions, we also note that this condition commonly

*Table 1.* The comparison of final test accuracy on the two datasets. The best results are **bolded**.

| Method | SVHN(%) | | | CIFAR-10(%) | | | CIFAR-100(%) | | |
|---|---|---|---|---|---|---|---|---|---|
| *Shards* (*M*,*S*) | (100, 2) | (100, 4) | (100, 8) | (100, 2) | (100, 4) | (100, 8) | (100, 2) | (100, 4) | (100, 8) |
| SemiFL | 88.36±1.11 | 93.33 ±0.48 | 92.88 ±2.33 | 66.34±3.21 | 76.68±3.20 | 81.80±2.79 | 37.73±0.86 | 42.68±2.12 | 47.87±1.66 |
| FedMatch | 73.50±3.11 | 73.57±1.34 | 73.66±1.15 | 45.43±2.48 | 45.04±0.47 | 45.03±1.57 | 16.89±2.97 | 17.05±2.94 | 17.05±3.45 |
| FedU | 83.78±1.87 | 86.50±1.85 | 86.60±1.74 | 61.51±1.03 | 61.62±3.12 | 65.74±1.66 | 30.83±2.78 | 32.23±1.77 | 33.57±2.09 |
| FedEMA | 84.40±0.46 | 86.62±0.89 | 85.77±1.68 | 65.08±2.55 | 68.23±1.81 | 66.80±0.43 | 29.48±2.43 | 32.86±2.65 | 33.62±3.08 |
| Orchestra | 86.57±1.78 | 86.99±1.70 | 87.59±3.20 | 68.57±2.55 | 69.09±1.98 | 69.03±2.35 | **39.93**±2.17 | 38.95±1.02 | 38.01±2.60 |
| (FL)$^2$ | 84.17±1.52 | 82.78±1.33 | 87.59±1.35 | 60.63±2.06 | 62.61±1.91 | 61.82±1.73 | 27.73±3.04 | 28.47±3.01 | 29.11±2.93 |
| FedFAME | 86.84±0.93 | 86.39±1.22 | 87.35±1.71 | 63.41±1.73 | 61.86±0.92 | 62.43±0.53 | 25.91±1.95 | 26.05±2.07 | 25.50±1.09 |
| pFedKnow | 56.88±2.73 | 59.80±2.06 | 59.58±1.33 | 59.93±1.75 | 60.03±1.52 | 60.57±1.36 | 19.55±0.75 | 19.77±1.03 | 18.80±1.60 |
| Ours | **94.52**±0.71 | **96.32**±0.93 | **96.53**±0.73 | **78.31**±1.32 | **82.36**±0.91 | **84.40**±0.72 | 39.09±3.12 | **43.56**±2.24 | **49.75**±2.31 |

holds according to Bayes' theorem. Specifically, Bayes' Theorem states that the posterior probability $P(y|x)$ is proportional to the generation probability $P(x|y)$, i.e., $P(y|x) \propto P(x|y)P(y)$. As a consequence, by considering $f(y|x)$ and $p(x)$ as the posterior probability and generation probability respectively, the condition of Theorem 6.2 is reasonable. In fact, our evaluation also verifies our theoretical analysis that our Bayesian-based approach outperforms simply averaging, as shown in Figure 5a.

# 7. Experiments

## 7.1. Setup

**Datasets and Models.** We consider three popular datasets in experiments, i.e., SVHN (Netzer et al., 2011), CIFAR-10 (Krizhevsky et al., 2009) and CIFAR-100 (Krizhevsky et al., 2009). which contains $10, 10, 100$ classes respectively. We use Wide ResNet28x2 (Zagoruyko & Komodakis, 2016) as our backbone model for CIFAR10 and SVHN datasets and Wide ResNet28x8 for CIFAR100 datasets throughout our experiments, by following the settings of existing SSFL works (Diao et al., 2022). The numbers of labeled data at the server for CIFAR10, SVHN, and CIFAR100 datasets are $500, 500, 2500$ respectively.

**Data Partition.** We adopt two Non-IID data partition methods: Shards (McMahan et al., 2017) and Dirichlet (Lin et al., 2020). In the Shards setting, the sorted samples are shuffled into $M * S$ shards, and assigned to $M$ clients randomly. Dirichlet distribution uses $\alpha$ to characterize the degree of heterogeneity. We set $\alpha$ of Dirichlet: $\{0.1, 1, 10\}$ and shards for each client: $\{2, 4, 8\}$.

**Baselines**. We compare our method against both SSFL and unsupervised FL methods. UFL includes FedU (Zhuang et al., 2021), FedEMA (Zhuang et al., 2022) and Orchestra (Lubana et al., 2022) while SSFL approaches contain FedMatch (Jeong et al., 2021), (FL)$^2$ (Lee et al., 2024), pFedKnow (Wang et al., 2023b), FedFAME (Malaviya et al., 2023), and SemiFL (Diao et al., 2022). We exclude the local-only method due to its deficiency.

**Implementation.** We implement the whole experiment in a simulation environment based on PyTorch 2.0 and 4 NVIDIA GeForce RTX 3090 GPUs. We use 100 clients in total and randomly choose 10% each round for local training. We set the local epoch to 5, batch size to 10, and learning rate to $3.0e - 2$. We employ SGD optimizer with the momentum of $0.9$ and weight decay of $5e - 4$ for all methods and datasets. The number of global communication rounds is 800. Each experiment is run 3 times and we take each run's final 10 rounds' accuracy to calculate the average value and standard variance. We set the threshold of our method to be $0.7$. For SemiFL and FedMatch, we adopt the same thresholds as leveraged in their original paper, i.e., $0.95$ for both SemiFl and for FedMatch.

## 7.2. Comparison with Baselines

**Shards-based NonIID.** Table 1 compares results with baselines in Shards-based Non-IID scenarios. Our proposed BSemiFL consistently achieves **best** performance in nearly all settings. For CIFAR-10 with 2 shards, BSemiFL achieves $78.31\%$ accuracy, outperforming Orchestra's $68.57\%$ by $9.8\%$, highlighting its effectiveness with highly non-IID data. However, on CIFAR-100 with 2 shards, Orchestra outperforms BSemiFL. Regarding the slightly lower average value compared to Orchestra, we believe this is because Orchestra is a federated self-supervised learning method. In this specific scenario, its better performance is mainly due to the fact that the client data distribution is highly concentrated in a few categories. This local consistency makes it easier for local clustering to capture clear category patterns. Since they focus on Step 2 of SSFL, Our method can be jointly used with them.

**Dirichlet-based NonIID.** Figure 4 compares our method with baselines in Dirichlet NonIID settings. Our method outperforms most baselines, achieving $76.71\%$ accuracy vs. SemiFL's $69.03\%$ ( $7\%$ lower). However, on CIFAR-10 with $\alpha = 10$, SemiFL slightly outperforms our method ($88.19\%$ vs. $87.27\%$). This is because SemiFL uses the global model to label local unlabeled data, performing better when the global model is well-trained (low NonIID).

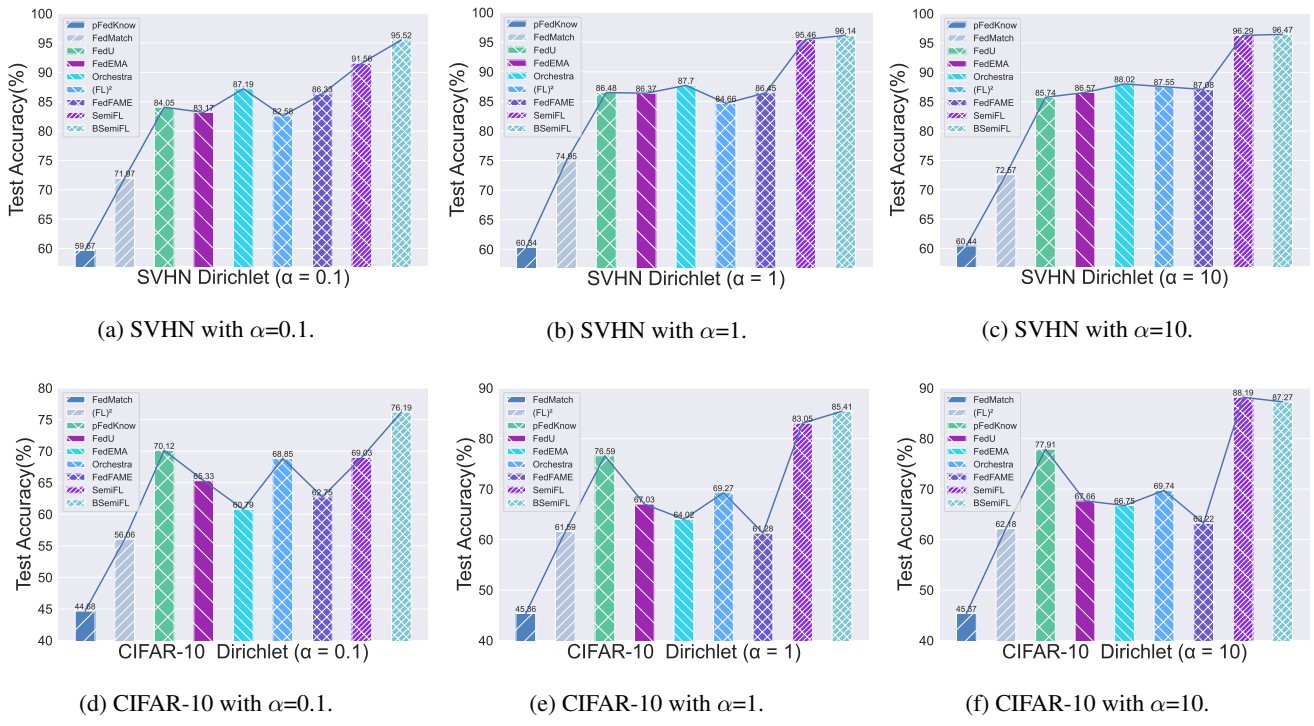

*Figure 4.* The comparison of final test accuracy on the NonIID setting of Dirichlet.

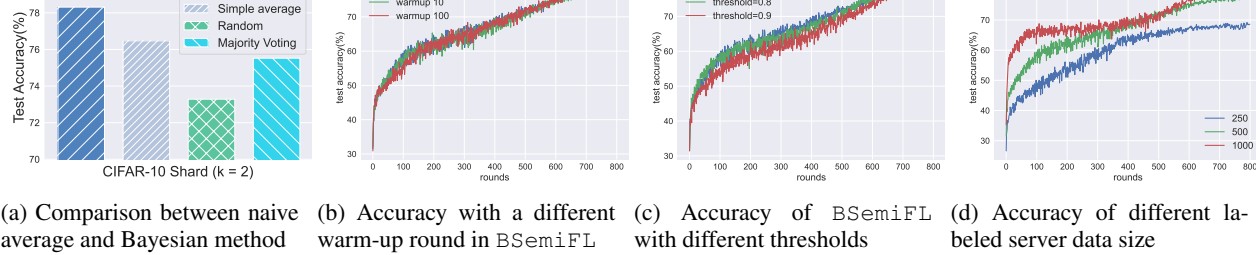

(a) Comparison between naive average and Bayesian method

(b) Accuracy with a different warm-up round in `BSemiFL`

(c) Accuracy of `BSemiFL` with different thresholds

(d) Accuracy of different labeled server data size

*Figure 5.* Results of ablation experiments.

However, SemiFL struggles with high NonIID, while our method remains robust.

### 7.3. Sensitivity Analysis and Ablation Study

**Impact of Ensemble Strategy.** Figure 5 shows ablation experiments on CIFAR-10 using shards(S=2) Non-IID. Figure 5a compares ensemble strategies: simple averaging, majority voting, random, and our Bayesian method. Simple average refers to assigning equal weights (0.5) to the results produced by the global model and the local model, and then computing a weighted sum. 'Random' refers to randomly assigning two weights to the results produced by the two models and then computing a weighted sum. 'Majority Vote' means that for a given unlabeled data point, we only

assign a pseudo-label if both the global model and the local model produce the same pseudo-label for that data point. Our Bayesian method achieves the best performance, indicating the importance of adaptively setting weights for local and global models.

**Impact of Warmup.** Figure 5b shows results for various warm-up rounds: 10, 20, and 100. Global accuracy differs minimally among methods. Accuracy is nearly identical at 78% for 10 and 20 rounds, but increases to about 79.5% at 100 rounds. This suggests a longer warm-up period can modestly enhance performance and overall accuracy.

**Impact of Threshold.** Figure 5c shows the impact of different threshold settings (0.7, 0.8, and 0.9) in `BSemiFL` for generating pseudo labels, with other parameters at default

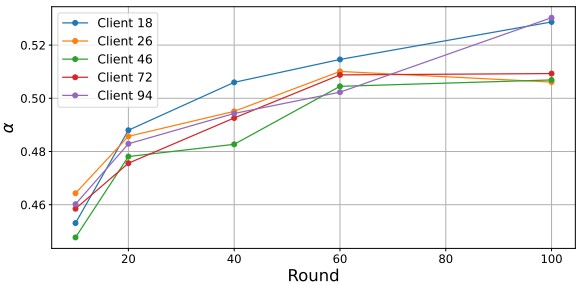

*Figure 6.* Value investigation of weight $\alpha$ among different clients.

values (e.g., 20 warm-up rounds). Results indicate minimal differences in final convergence accuracy, all around 78%. This robustness to threshold variations ensures consistent performance and reduces manual hyperparameter tuning costs, enhancing practicality for real-world use.

**Impact of Labeled Dataset Size.** Figure 5d shows the impact of labeled data size (250, 500, 1000 samples) on test accuracy. Accuracy rises significantly with more labeled data, but gains diminish as the dataset grows. Increasing from 250 to 500 samples boosts accuracy from 68% to 78% (a 10% gain), while increasing from 500 to 1000 samples only improves accuracy from 78% to 81% (a 3% gain). This indicates diminishing returns, guiding efficient resource allocation for labeling efforts.

**Robustness to Data Heterogeneity.** Comparing results of different degrees of statistical heterogeneity exhibited in Table 1 and Figure 4, we can find that BSemiFL presents greater advantages over baselines when the data is more heterogeneous. For example, our method outperforms SemiFL by 6% when the shard $S = 2$ while achieving an improvement by 4% when the shard $S = 8$ for SVHN dataset. Besides, it can also be observed that our method achieves similar performance under different NonIID settings. For example, the performance of our method ranges from 95.52% to 96.47% when $\alpha$ ranges from 0.1 to 10. As a comparison, the performance of SemiFL ranges from 91.56% to 96.29%. The results indicate that *our method is more robust to the degree of NonIID*.

**Investigation of Weight $\alpha$.** To investigate the values of weight $\alpha$ during training process, we add the following experiment. When each client is selected for training, we record the average value of $\alpha$ for all data points in its local dataset. We here present the corresponding values for five randomly selected clients, as shown in Figure 6. As can be seen, in the initial rounds, the local model achieves higher accuracy on the local data compared to the global model, resulting in a relatively smaller weight $\alpha$ for the global model. In later rounds, the values of $\alpha$ gradually stabilize around 0.5, which aligns with the gradual improvement in the accuracy of the global model.

## 8. Conclusion

Re-labeling local data is crucial in Semi-supervised Federated Learning. This paper theoretically and empirically identifies that using only the global or local model limits performance due to distribution gaps and local errors. Considering these limits, this paper proposes BSemiFL, a method that collaboratively re-labels local data using a weighted ensemble of both models, with weights determined by a Bayesian approach. Our analysis shows that the re-labeling loss of BSemiFL is theoretically smaller than the maximum individual loss of either the global or the local model. Besides, the labeling loss of the Bayesian-based ensemble of the global and local models is also theoretically smaller than simply averaging them. Empirically demonstrate that BSemiFL improves the performance by up to 9.8% as compared to state-of-the-art methods.

## Acknowledgement

This work is supported by the National Key Research and Development Program of China under grant 2024YFC3307900; the National Natural Science Foundation of China under grants 62302184, 62376103, 62436003, 62206102, and 62402331; Major Science and Technology Project of Hubei Province under grant 2024BAA008; Hubei Science and Technology Talent Service Project under grant 2024DJC078; and Ant Group through CCF-Ant Research Fund. This work is also supported by two grants from the Research Grants Council of the Hong Kong Special Administrative Region, China (Project No. PolyU15222621, PolyU15225023), and is supported by the Fundamental Research Funds for the Central Universities (Grant No. YJ202429). Besides, this work is supported by the National Research Foundation, Singapore, and Cyber Security Agency of Singapore under its National Cybersecurity R&D Programme and CyberSG R&D Cyber Research Programme Office. Any opinions, findings and conclusions or recommendations expressed in these materials are those of the author(s) and do not reflect the views of National Research Foundation, Singapore, Cyber Security Agency of Singapore as well as CyberSG R&D Programme Office, Singapore.

## Impact Statement

Our approach enables learning from mostly unlabelled data, promoting resource efficiency. In healthcare and education, it accelerates knowledge and service improvements, offering high-quality services to remote or under-resourced areas, thus fostering fairness. Yet, ensuring algorithmic fairness and transparency remains a challenge to avoid increasing social inequalities.

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

# A. Proofs

**Preliminary of Theories**. Our following analysis relies on the following theorem which bounds the transferring error by adapting the model of one domain to another domain.

**Proposition A.1.** *(Theorem 1 in (Ben-David et al., 2010)). Considering the distributions $p_s$ and $p_t$, for every hypothesis $f \in \mathcal{F}$ and any constant $\delta \in (0,1)$, with probability at least $1 - \delta$, there exists*

$$L_{p_t}(f) \le L_{p_s}(f) + \frac{1}{2} d_{\mathcal{F}\Delta\mathcal{F}}(p_s, p_t) + \lambda, \tag{18}$$

*where $d_{\mathcal{F}\Delta\mathcal{F}}(p_s, p_t)$ measures the distance between the distribution $p_s$ and $p_t$. $L_{p_s}(f)$ denotes the expected labeling error of the hypothesis $f$ over the data distribution $p_s$, i.e., $L_{p_s}(f) = \mathbb{E}_{x\sim p_s} l(f, x)$, and $L_{p_t}(f)$ has a similar meaning to $L_{p_s}(f)$. $\lambda = L_{p_s}(f_*) + L_{p_t}(f_*)$ with $f_* = \arg\min_{f\in\mathcal{F}} L_{p_s}(f) + L_{p_t}(f)$.*

**Theorem A.2.** *Denote the data distribution of each client $m$ by $p_m$, the empirical distribution of the global dataset by $\hat{p}$, and the real distribution of the global data by $p$. Given constants $0 < \delta \le 1$ and $\sigma > 0$, with the probability at least $1 - \delta$, the expected labeling error $L_{p_m}(f_{\hat{p}})$ of the global hypothesis $f_{\hat{p}}$ over the local data distribution $p_m$ is bound by:*

$$L_{p_m}(f_{\hat{p}}) \le L_{\hat{p}}(f_{\hat{p}}) + \sqrt{\frac{\log\frac{2}{\delta}}{2S}} + \frac{1}{2} d_{\mathcal{F}\Delta\mathcal{F}}(p_m, p) + \lambda_g, \tag{19}$$

*where $S = \sum_{m=1}^{M} S_m + N_s$ denotes the total data size of the global dataset, and $\lambda_g = L_p(f_*) + L_{p_m}(f_*)$ with $f_* = \arg\min_{f\in\mathcal{F}} L_p(f) + L_{p_m}(f)$.*

*Proof.* For convenience of expression, we denote the hypothesis $f(x, \mathbf{w})$ by $f$ and denote $\arg\min_{f\in\mathcal{F}} L_{\hat{p}}(f)$ by $f_{\hat{p}}$. According to the Hoeffding inequality, with probability at least $1 - \delta$, there exists

$$L_p(f_{\hat{p}}) \le L_{\hat{p}}(f_{\hat{p}}) + \sqrt{\frac{\log\frac{2}{\delta}}{2S}}. \tag{20}$$

Then, by denoting the distribution of the local data in client $m$ as $p_m$, based on the Proposition 1, we have

$$L_{p_m}(f_{\hat{p}}) \le L_p(f_{\hat{p}}) + \frac{1}{2} d_{\mathcal{F}\Delta\mathcal{F}}(p_m, p) + \lambda_g, \tag{21}$$

Jointly considering (20) and (22) together, we have

$$L_{p_m}(f_{\hat{p}}) \le L_{\hat{p}}(f_{\hat{p}}) + \sqrt{\frac{\log\frac{2}{\delta}}{2S}} + \frac{1}{2} d_{\mathcal{F}\Delta\mathcal{F}}(p_m, p) + \lambda_g, \tag{22}$$

which completes the proof. $\qquad\square$

**Theorem A.3.** *Denote the hypothesis trained on the empirical distribution of the local dataset $\hat{p}_m$ by $f_{\hat{p}_m}$ and the hypothesis trained on the empirical distribution of the global dataset $\hat{p}$ by $f_{\hat{p}}$. For any local data sample $x \sim p_m$, considering the Cross-Entropy loss function is adopted, then the loss of the ensemble between the global and local model is less than the maximum individual loss of either the global or local model, i.e.,*

$$L(\alpha f_{\hat{p}} + (1 - \alpha) f_{\hat{p}_m}) \le \max\big(L(f_{\hat{p}}), L(f_{\hat{p}_m})\big). \tag{23}$$

*Proof.* Considering the convexity of the Cross-Entropy loss function in terms of the prediction, for any $x \sim p_m$, we have

$$\begin{aligned}
L(\alpha f_{\hat{p}} + (1 - \alpha) f_{\hat{p}_m}) &\le \alpha L(f_{\hat{p}}) + (1 - \alpha) L(f_{\hat{p}_m}) \\
&\le \max\big(L(f_{\hat{p}}), L(f_{\hat{p}_m})\big).
\end{aligned} \tag{24}$$

$$\square$$

**Theorem A.4.** *Consider the same notations of distribution and hypothesis defined in previous theorems. For each local sample $x \sim p_m$ with the ground-truth label $y = k$, considering the Cross-Entropy loss function is adopted, then under the condition that $f_{\hat{p}}(k|x) > f_{\hat{p}_m}(k|x)$ when $\hat{p}(x) > \hat{p}_m(x)$ and vice versa, the loss of Bayesian-based model ensemble is smaller than the direct averaging:*

$$L(\alpha f_{\hat{p}} + (1 - \alpha)f_{\hat{p}_m}) \leq L(\frac{f_{\hat{p}} + f_{\hat{p}_m}}{2}). \tag{25}$$

*Proof.* According to the definition of the Cross-Entropy loss, we have

$$
\begin{aligned}
&L(\alpha f_{\hat{p}} + (1 - \alpha)f_{\hat{p}_m}) \\
&= -\log(\alpha(x) f_{\hat{p}}(k|x) + (1 - \alpha(x)) f_{\hat{p}_m}(k|x)) \\
&= -\log \left( \frac{\hat{p}(x)}{\hat{p}(x) + \hat{p}_m(x)} f_{\hat{p}}(k|x) + \frac{\hat{p}_m(x)}{\hat{p}(x) + \hat{p}_m(x)} f_{\hat{p}_m}(k|x) \right) \\
&= -\log \left( \frac{f_{\hat{p}}(k|x) + f_{\hat{p}_m}(k|x)}{2} + \frac{(\hat{p}(x) - \hat{p}_m(x))}{2(\hat{p}(x) + \hat{p}_m(x))} \left( f_{\hat{p}}(k|x) - f_{\hat{p}_m}(k|x) \right) \right) \\
&\leq -\log \left( \frac{f_{\hat{p}}(k|x) + f_{\hat{p}_m}(k|x)}{2} \right) \\
&= L(\frac{f_{\hat{p}} + f_{\hat{p}_m}}{2}), \tag{26}
\end{aligned}
$$

where the last inequality is derived based on the condition that $(\hat{p}(x) - \hat{p}_m(x))(f_{\hat{p}}(k|x) - f_{\hat{p}_m}(k|x))$ is always positive. $\square$

## B. Algorithm Workflow

The workflow is presented in Algorithm 1. In lines 4-5, the server first trains the global model on the labeled dataset and then calculates the sum of sampling probability. Next, in lines 6-8, the server distributes the global model and probability sum to all selected clients. Each client mainly conducts two main steps as follows.

• **Labeling local data (lines 14-21):**

  1. Each client $m$ calculates the closeness of the global model in lines 16-17;

  2. Each client $m$ calculates the closeness of the local model in lines 18-19;

  3. Each client $m$ applies an ensemble of the global and local model to construct the labeled dataset in lines 20-22.

• **Local training (lines 24-25):**

  1. Each client $m$ performs local training with labeled dataset for $E$ local epochs;

  2. Each client $m$ pushes the local model to the server.

Finally, in lines 9-11, the server aggregates the local models of all participated clients into the global model.

---

**Algorithm 1** Algorithm workflow of `BSemiFL`

---

**Input** : $T$: round; $M$: client number; $\eta$: learning rate;

1 Initialize the parameter $\mathbf{w}_0$;

  **In server**:

  **for** $t = 1$ **to** $T$ **do**

2     Fine-tune the global model $\mathbf{w}_t$ on the dataset $\mathbb{S}$;

     Calculate the sum of sampling probability of the server dataset $\mathbb{S}$ for all $K$ classes: $Q_s^k = \sum_{i=1}^{N_s} f_{\hat{p}}(k|x_i, \mathbf{w}_t)$;

     Randomly select $M_t$ clients;

     **for** *each selected client $m$* **in parallel do**

3         Send the global model $w^t$ and probability vector $Q_s$;

         Receive the local model $\mathbf{w}_t^m$;

4     **end**

5     Aggregate local models: $\mathbf{w}_{t+1} = \sum_{m=1}^{M_t} \frac{S_m}{S^t} \mathbf{w}_t^m$;

6 **end**

7 **In client** $m$:

  **for** *each sample $x_i^m \in \mathbb{U}_m$* **do**

8     Calculate the output of the global model $f_{\hat{p}}(x_i^m, \mathbf{w}_t)$ and of the local model $f_{\hat{p}_m}(x_i^m, \mathbf{w}_{\tilde{t}^m})$;

     Calculate the conditional probability of the global model: $\hat{p}(x_i^m|k) = \frac{f_{\hat{p}}(k|x_i^m, \mathbf{w}_t)}{f_{\hat{p}}(k|x_i^m, \mathbf{w}_t) + Q_s^k}$ for all classes $k = 1, \ldots, K$;

     Calculate the closeness of the global model: $\hat{p}(x_i^m) = \sum_{k=1}^{K} \hat{p}(x_i^m|k)\hat{p}_s(k)$;

     Calculate the conditional probability of the local model: $\hat{p}_m(x_i^m|k) = \frac{f_{\hat{p}_m}(k|x_i^m, \mathbf{w}_t^m)}{\sum_{i=1}^{S_m} f_{\hat{p}_m}(k|x_i, \mathbf{w}_t^m)}$ for all $K$ classes;

     Calculate the closeness of the local model: $\hat{p}_m(x_i^m) = \sum_{k=1}^{K} \hat{p}_m(x_i^m|k)\hat{p}_m(k)$;

     Normalizes the weights: $\alpha_i^m = \frac{\hat{p}(x_i^m)}{\hat{p}(x_i^m) + \hat{p}_m(x_i^m)}, \quad 1 - \alpha_i^m = \frac{\hat{p}_m(x_i^m)}{\hat{p}(x_i^m) + \hat{p}_m(x_i^m)}$;

     Apply an ensemble of the global and local model: $\hat{y}_i^m = \alpha_i^m f_{\hat{p}}(x_i^m, \mathbf{w}_t) + (1 - \alpha_i^m)f_{\hat{p}_m}(x_i^m, \mathbf{w}_t^m)$;

     Constructs the dataset $\mathbb{S}_m$ by conducting (2);

9 **end**

10 Update local model $\mathbf{w}_t^m$ for $E$ local epochs on the labeled dataset $\mathbb{S}_m$: $\mathbf{w}_t^m = \mathbf{w}_t^m - \eta \nabla_{\mathbf{w}_t^m} L_m$;

  Send the model $\mathbf{w}_t^m$ to the server;

---

