# OpenReview forum: "BSemiFL: Semi-supervised Federated Learning via a Bayesian Approach"
_ICML.cc/2025/Conference — ICML 2025 poster_

### Official Review · Reviewer_YYii · 2025-03-05

**Overall Recommendation:** 3

**Summary:**

1. This paper proposes BSemiFL, a federated semi-supervised learning framework. BSemiFL theoretically demonstrates why solely relying on either the global model or the local model for labeling local data is suboptimal.
2. The authors employ a Bayesian approach to evaluate the proximity of local and global models to the samples, and then dynamically weight their contributions to pseudo-label prediction based on their inferred relevance.

**Claims And Evidence:**

Yes.

**Essential References Not Discussed:**

FedDB[3] also employs Bayesian analysis in the federated semi-supervised learning setting.

Compared to this work: FedDB explicitly models class prior bias arising from imbalanced data, aiming to mitigate its impact throughout the training process. In contrast, BSemiFL focuses solely on weighting the predictions of global and local models, without addressing the internal class bias of the model or adjusting the prior distribution within the predictions. Moreover, the analysis in BSemiFL is limited to its aggregation strategy and does not provide a comparative evaluation against the debiasing strategy proposed in FedDB.

**Experimental Designs Or Analyses:**

Yes, I have checked it. The main contribution of this paper lies in its ensemble strategy. In the experimental section, the authors compare their proposed strategy with other ensemble approaches and demonstrate its superior performance.

**Methods And Evaluation Criteria:**

Yes, it is practically meaningful. BSemiFL enables the dynamic adjustment of the contributions from local and global models in pseudo-label generation, which provides a certain contribution to addressing data heterogeneity in federated semi-supervised learning. However, its novelty is somewhat limited.

**Other Comments Or Suggestions:**

This paper ignores the cumulative amplification of pseudo-label errors across multiple rounds:The theoretical analysis focuses on a single re-labeling step, overlooking the long-term accumulation and propagation of pseudo-label noise through iterative training rounds, which can degrade model performance over time.

**Other Strengths And Weaknesses:**

Strengths:
1. The writing is clear and fluent.
2. One of the main contributions of this paper lies in its ensemble strategy. In the experimental section, the authors appropriately validate this contribution by comparing their method with other ensemble strategies.

Weaknesses:
1. The theoretical analysis has certain limitations. Specifically, the paper assumes that the local data distribution is known. However, in semi-supervised settings, the local distribution is typically unknown. Although pseudo-labels can assist in estimating the distribution, they are prone to errors. The impact of pseudo-label noise is not considered in the knowledge derived from the local models.

2. The novelty is somewhat limited. The limitations of local and global models under data heterogeneity have become widely recognized in federated learning. Although this work attempts to extend such insights to FSSL, the characteristics of FSSL are not sufficiently reflected—particularly due to the assumption of known local data distributions.

**Questions For Authors:**

Currently, many foundation models, such as CLIP, have been introduced into FL and FSSL, where their prior knowledge helps alleviate the problem of client data imbalance. In this context, how do you evaluate the improvements brought by the method proposed in this paper compared to foundation models in pseudo-label generation? Furthermore, do you think the theoretical analysis in this paper could be extended to incorporate considerations related to foundation models?

 [1] Cho, Y.J., Joshi, G. and Dimitriadis, D., 2023. Local or global: Selective knowledge assimilation for federated learning with limited labels. In Proceedings of the IEEE/CVF International Conference on Computer Vision.
 [2] Liu, Y., Wu, H. and Qin, J., 2024, March. Fedcd: Federated semi-supervised learning with class awareness balance via dual teachers. In Proceedings of the AAAI Conference on Artificial Intelligence.
 [3] Zhu, G., Liu, X., Wu, X., Tang, S., Tang, C., Niu, J. and Su, H., 2024, January. Estimating before debiasing: A Bayesian approach to detaching prior bias in federated semi-supervised learning. In Proceedings of the International Joint Conference on Artificial Intelligence.

**Relation To Broader Scientific Literature:**

Previous studies in the field of FSSL have also utilized both local and global models for pseudo-label generation, such as [1,2]. However, these works did not provide theoretical analysis to demonstrate the limitations of relying on a single model.

**Theoretical Claims:**

Starting from the empirical distributions of the global dataset and local datasets, it theoretically proven that the ensemble loss between the global model and local models is lower than the maximum individual loss of either the global model or any local model.

---

> ### Author Rebuttal · Authors · 2025-04-01
>
> Thanks a lot for the reviewing.
>
> ```
> Q1.FedDB[3] also employs Bayesian in FSSL. FedDB explicitly and mitigates class prior bias from imbalanced data. BSemiFL does not address internal class bias. Additionally, BSemiFL’s analysis is confined to its aggregation strategy and lacks a comparative evaluation.
> ```
> Although [3] also employs Bayesian methods, the scenarios in which the two methods are applicable are entirely different. The scenario considered by [3] is "labels at client," which may not be suitable for our "labels at server" scenario. The core step of [3] involves using a single model to annotate local data and suppress the prediction probabilities of the majority class to correct annotation errors of that single model.
> - However, this method requires calculating the prior probability of the majority class, which relies heavily on labeled data from the "labels at client" setting. When there is no labeled data locally, an initial miscalculation of the prior probability will exacerbate errors during the suppression process, and there would be no way to correct these priors based on labeled data later on. In contrast, our proposed method does not depend on local labeled datasets but instead leverages a Bayesian weighting approach between the local and global models to jointly correct prediction probabilities.
> - The fundamental idea of [3] is a weighted inference method for addressing class imbalance, which has inherent limitations. For example, in a three-class classification task, assuming class C1 is the majority class and classes C2 and C3 are minority classes with equal sample sizes, the trained model might tend to classify C2 and C3 as C1. Although [3] penalizes the majority class, due to insufficient training on minority classes, it still cannot distinguish between C2 and C3, leading to misclassification. Therefore, widely used methods in the field of class imbalance are oversampling rather than weighting. Our method adaptively assigns the labeling of the majority class to the local model and the labeling of minority classes to the global model, thereby avoiding such errors.
>
> Cifar10
>
> | |dir d=0.1| shard k=2|
> |-|-|-|
> |FedDB|66.78|66.14|
> |Our|76.19|78.31|
>
> Thus, [3] and our method apply to different scenarios. Indeed, combining the error correction of a single model from [3] with the correction of integrated annotations from two models in our method could potentially lead to further improvements and broader applicability across various scenarios.
> ```
> Q2. Theories have limitations: the paper assumes a known local distribution. using pseudo-labels to estimate the distribution has noise.
> ```
> Although the pseudo-label estimation distribution contains some noise, as training progresses, the model's accuracy improves, and this estimation error gradually decreases. Moreover, while the theorem has certain limitations, it still demonstrates the effectiveness of the method to a certain extent. It shows that under approximate conditions, the method outperforms using a single model or a simple average of two models.
>
> ```
> Q3：Novelty is somewhat limited. The limitations of data heterogeneity have been recognized in FL. Although this work seeks to extend it to FSSL, the characteristics are not sufficiently reflected.
> ```
> This paper investigates multiple characteristics of FSSL, including but not limited to:
> - The first systematic exploration of the advantages and disadvantages of local-only and global-only labeling methods, as well as the underlying reasons, which has not been addressed in traditional FL.
> - The first proposal of an adaptive weighting integration of global and local models tailored to the labeling needs in FSSL, a unique feature in FSSL.
> - Our theoretical analysis specifically targets the accuracy of labeling in FSSL.
> Additionally, we adopt an approximation of pseudo-label distributions and do not require prior knowledge of the local data distribution.
>
> ```
> Q4. The paper ignores the cumulative amplification of pseudo-label errors across multiple rounds. The theories focus on a single re-labeling step, overlooking the long-term accumulation of noise.
> ```
> Our method involves labeling in conjunction with the global model, which can be refined using the global labeled dataset, thereby reducing such noise. This is also reflected in SemiFL, where using only the global model can gradually lead to better performance without amplifying cumulative errors.
>
> ```
> Q5.Many big models were used in FL. Their knowledge helps alleviate the imbalance. How to evaluate improvements of the method compared to big models in pseudo-label generation? Can theories incorporate big models?
> ```
> Our method can be combined with foundational models to a certain extent. For example, we can treat the foundational model as the global model and integrate it with the local model to label local data. Considering the differences between foundational models and global models, new theoretical and experimental analyses may be required.

---

### Official Review · Reviewer_uy97 · 2025-03-11

**Overall Recommendation:** 5

**Summary:**

This paper focus on the semi-supervised scenarios in Federated Learning. This paper delves deeply into the performance dominance and limitations of the global and local models for relabeling the local data from both theoretical and empirical perspectives. Then, they propose a novel method which re-labels the local data through the collaboration between the local and global model in a Bayesian approach. Established theories demonstrate the effectiveness of their proposed method. Experimental results also show that their method greatly improves the performance.

## update after rebuttal
After the Reviewer-author discussion phase, I maintain my score and explicitly support acceptance.

**Claims And Evidence:**

This paper presents the challenge of using single model from both theoretical and empirical perspectives.

**Essential References Not Discussed:**

The references are sufficient.

**Experimental Designs Or Analyses:**

The experimental designs are comprehensive and reasonable to evaluate the proposed method. Specifically, the experimental datasets and tasks are widely adopted in this SSFL area, which meets the criteria. Besides, the compared baselines are sufficient enough to demonstrate the effectiveness of the proposed method.

**Methods And Evaluation Criteria:**

The method is well elaborated and has strong motivations. The designed evaluations are sufficient to verify that the method can solve their identified problem, meeting the criteria.

**Other Comments Or Suggestions:**

No.

**Other Strengths And Weaknesses:**

Strengths:

1.	The empirical and theoretical analysis about the limits and benefits of using single models are sound and clearly present the motivations.

2.	The proposed method of using Bayesian-based ensemble of the global and local model to relabel the local data is interesting and innovative.

3.	The theoretical validation of the performance is solid and guarantees the effectiveness of the proposed method.

4.	The presentation is good and clear, making the paper easy to follow.

5.	The experiments compare their method with many baselines over different datasets and NonIID settings, which are sufficient.

Weakness and Concerns:

1.	The details of the designed ensemble strategies in Figure 5(a) are not specified. It seems that these methods are designed by this paper itself. However, the details are not clear. For example, what’s the ,meaning of the majority voting? In my opinion, there are only two models, which has the concept of “majority”.

2.	I'm curious as to why your method performs worse than Orchestra when it comes to Cifar100 (100, 2)?

**Questions For Authors:**

See above.

**Relation To Broader Scientific Literature:**

This paper focuses on the integration of semi-supervised learning and federated learning, with particular attention to the issue of inaccurate labeling caused by NonIID. Therefore, it is related to works on semi-supervised learning in centralized scenarios [1] and efforts addressing NonIID in federated learning [2].
[1] Z. Zhu et al. The rich get richer: Disparate impact of semi-supervised learning. ICLR 2022.
[2] SP Karimireddy et al. Scaffold: Stochastic controlled averaging for federated learning. ICML 2020.

**Theoretical Claims:**

The theories clearly illustrate the limits of using single models and guarantee the effectiveness of the proposed method. The proofs are clear and correct.

---

> ### Author Rebuttal · Authors · 2025-04-01
>
> Thank you very much for your reviewing and valuable suggestions.
>
> ```
> Q1：The details of the designed ensemble strategies in Figure 5(a) are not specified. It seems that these methods are designed by this paper itself. However, the details are not clear. For example, what’s the ,meaning of the majority voting? In my opinion, there are only two models, which has the concept of “majority”.
> ```
> We apologize for the lack of clarity in some of our descriptions:
>
> In Figure 5(a):
> - **Simple average** refers to assigning equal weights (0.5) to the results produced by the global model and the local model, and then computing a weighted sum.
> - **Random** refers to randomly assigning two weights to the results produced by the two models and then computing a weighted sum.
> - **Majority Vote** means that for a given unlabeled data point, we only assign a pseudo-label if both the global model and the local model produce the same pseudo-label for that data point.
>
>
> ```
> Q2：I'm curious as to why your method performs worse than Orchestra when it comes to Cifar100 (100, 2)?
> ```
> Thank you for your feedback.
>
> First, our method achieves a performance of **39.09±3.12** on CIFAR100 (100, 2), while Orchestra achieves **39.93±2.17**. These results are based on multiple experiments, so the performance of our method is not inferior to that of Orchestra. Regarding the slightly lower average value compared to Orchestra, we believe this is because Orchestra is a federated self-supervised learning method. In this specific scenario, its better performance is mainly due to the fact that the client data distribution is highly concentrated in a few categories. This local consistency makes it easier for local clustering to capture clear category patterns.
>
> In contrast, our method does not rely on clustering but instead leverages the collaboration between the global model and the local model to assign pseudo-labels. Given the higher complexity of CIFAR100 images compared to other datasets, this leads to slightly higher fluctuations in our method compared to Orchestra.

---

### Official Review · Reviewer_kFQg · 2025-03-12

**Overall Recommendation:** 4

**Summary:**

This paper aims to solve the problem in Semi-supervised Federated Learning where the local data labels are absent in clients. They first theoretically and empirically demonstrate that the limitations and benefits of local model and the global model for relabeling the local data. They propose a new method which re-labels the local data through the collaboration between the local and global model in a Bayesian approach. Finally, they theoretically empirically demonstrate the effectiveness of their proposed method.

**Claims And Evidence:**

This paper claims that using a single model (e.g., a local or global model) for relabeling the local data will lead to poor generalization or personalization. They verify this from both theoretical and empirical perspectives. Besides, they claim that Bayesian based ensemble can solve this problem. Their established theories demonstrate that the achieved performance outperforms a single global or local model or their simply average.

**Essential References Not Discussed:**

The references are sufficient..

**Experimental Designs Or Analyses:**

They use SVHN, CIFAR10, and CIFAR100 as the evaluation dataset and Resnets as  models, which is a generalized adopted benchmark. They compare their proposed method with 8 baselines including both SSFL and Unsupervised FL methods, which are sufficient enough to verify the effectiveness. The analysis also explains the reason for the phenomenon.

**Methods And Evaluation Criteria:**

They propose using the weighted ensemble of the local and global model to re-label the local data, where the weights are calculated by the Bayesian approach. They evaluate the proposed method by comparing 8 baselines over both shard and Dirichlet based NonIID distribution settings, which are sufficient to me. Besides, the ablation study also verify the effectiveness of the Bayesian based ensemble.

**Other Comments Or Suggestions:**

No

**Other Strengths And Weaknesses:**

Strengths:

1.	The idea of using Bayesian based method is interesting and novel. The design of weighted ensemble to leverage both benefits of two models makes sense.

2.	The motivation is clear. The analysis reveal the underlying principles behind the problem using a single global or local model.

3.	The techniques are sound and the established theories are solid, which corresponds to the empirical observations.

4.	The paper is well organized and the writing is good.

5.	The evaluations are sufficient which many recent baselines and various settings.

Weakness:

1.	The proposed method requires more interaction items except of the models between the clients and the server. The risk of the privacy leakage is not discussed.

2.	There exists a SSFL method using Bayesian approach [1]. What’s the difference between your proposed method their method?

[1] Estimating before Debiasing: A Bayesian Approach to Detaching Prior Bias in Federated Semi-Supervised Learning. IJCAI 2024.

**Questions For Authors:**

See concerns.

**Relation To Broader Scientific Literature:**

The field of this paper focus on is de-facto a combination of the semi-supervised learning and federated learning. Although the proposed method achieves great effectiveness for SSFL, they only adopt a basic semi-supervised learning method in FL. In fact, more recent advanced SSL methods [1] can considered.

[1] BEM: Balanced and Entropy-Based Mix for Long-Tailed Semi-Supervised Learning. CVPR 2024.

**Theoretical Claims:**

Their theories can be divided into two main parts. The first part includes Theorem 4.1 and 4.2, which claim that using a single model (e.g., a local or global model) either cannot achieve generalization or cannot fill the distribution gap. The second part includes Theorem 6.1 and 6.2, which claim that their proposed method outperforms using a single model or the simple average of two models. The claims are consistent with the empirical observation and the proofs are correct.

---

> ### Author Rebuttal · Authors · 2025-04-01
>
> Thank you very much for your reviewing and valuable suggestions.
> ```
> Q1：more recent advanced SSL methods [1] can considered.
> [1] BEM: Balanced and Entropy-Based Mix for Long-Tailed Semi-Supervised Learning. CVPR 2024.
> ```
> BEM [1] mainly proposes a novel hybrid method for rebalancing the class distribution in terms of data quantity and uncertainty. This method can be integrated into our proposed approach.
>
> ```
> Q2：The proposed method requires more interaction items except of the models between the clients and the server. The risk of the privacy leakage is not discussed.
> ```
> Although our method requires more interaction terms, it does not actually increase the risk of privacy leakage. Compared to traditional FL methods, as shown in line 681, our method only needs to additionally transmit the K-dimensional probability distribution vector $Q_s=[Q_s^1,...,Q_s^K]$ of the server-side public dataset categories on the global model. In the SSFL scenario, the data on the server side is generally non-private. Therefore, in practice, our method does not result in additional client privacy leakage compared to traditional FL methods.
>
> ```
> Q3：There exists a SSFL method using Bayesian approach [1]. What’s the difference between your proposed method their method?
> [1] Estimating before Debiasing: A Bayesian Approach to Detaching Prior Bias in Federated Semi-Supervised Learning. IJCAI 2024.
> ```
> First, the scenario of our method differs from that of the referenced article:
>
> In our scenario, we assume that clients only have unlabeled data, while the server has a small amount of labeled data. In contrast, in [1], the scenario assumes that clients possess both supervised and unsupervised data, while the server does not have any data.
>
> Second, regarding the application of Bayes' theorem:
>
> 1.
> Our method leverages Bayes' theorem to assign pseudo-labels to local data based on the knowledge of both the local model and the global model. Specifically:
> We first fine-tune the global model on the server using the labeled data available on the server:
>    $ Q_s^k = \sum_{i=1}^{N_s} f_{\hat{p}}(k|x_i, w_t) $
>
> and then compute the probability distribution of the server-side data on the global model.
> This parameter, along with the global model, is distributed to the clients. Subsequently, we calculate:
>
>    $$
>    \hat{p}(x_i^m|k) = \frac{f_{\hat{p}}(k|x_i^m, w_t)}{f_{\hat{p}}(k|x_i^m,w_t) + Q_s^k}
>    $$
> For each data point in the local client, we compute:
>
>    $$
>    \hat{p}(x_i^m) = \sum_{k=1}^K \hat{p}(x_i^m|k)\hat{p}_s(k).
>    $$
>
> Similarly, in the local model, we compute the empirical distribution (using pseudo-labels from the previous round) and the probability distribution of the local data:
>
>    $ Q_m^k = \sum_{i=1}^{S_m} f_{\hat{p}_m}(k|x_i,w_t^m) $.
>
>    Then, we calculate$A=\sum_{k=1}^K f_{\hat{p}_m}(k|x_i^m, w_t^m)\hat{p}_m(k)$,
>
> and $B=(f_{\hat{p}_m}(k|x_i^m, w_t^m)+Q_m^k)$
> to obtain $\hat{p}_m(x_i^m)=A/B$
>
> Next, for each local data point, we obtain the confidence levels of the global model and the local model as:
>    $$
>    \alpha_i^{m} = \frac{\hat{p}(x_i^m)}{\hat{p}(x_i^m)+\hat{p}_m(x_i^m)}, \quad 1-\alpha_i^{m} = \frac{\hat{p}_m(x_i^m)}{\hat{p}(x_i^m)+\hat{p}_m(x_i^m)}.
>    $$
>
> Finally, by denoting $C=f_{\hat{p}}(x_i^m, w_t)$, and $D=f_{\hat{p}_m}(x_i^m, w_t^m)$, we derive the final probability distribution:
>
>    $$
>    \hat{y}_i^m =  \alpha_i^m C+ (1-\alpha_i^m) D
>    $$
>
> 2.
> In contrast, in FedDB, the use of Bayes' theorem occurs during local training, where the knowledge of the local model is utilized to optimize pseudo-labeling via APP-U (the Average Prediction Probability of Unlabeled Data), thereby reducing label prior bias. Specifically, it is argued that the imbalance in the local dataset introduces bias when the local model assigns pseudo-labels to unlabeled data, favoring classes with more data. To address this issue, the authors propose applying Bayes' theorem to rewrite:
>    $$
>    p_s(y|x) = \frac{e^{z(x)[y]}}{\sum_{k=1}^{K} e^{z(x)[k]}}
>    $$
>    as:
>    $$
>    p_s(y|x) = \frac{p_s(y) p_s(x|y)}{\sum_{k=1}^{K} p_s(k) p_s(x|k)}.
>    $$
> To mitigate the imbalance issue, a regularization-like term is introduced, yielding:
>    $$
>    \hat{p} = \frac{p(y|x)/\overline{p}}{\sum_{k=1}^{K} {p(k|x)}/\overline{p}_k},
>    $$
>
> Thus, our method primarily uses Bayes' formula to combine the knowledge of the global model and the local model through pseudo-labeling, whereas the work in [1] focuses on addressing data imbalance issues.

---

> > ### Comment · Reviewer_kFQg · 2025-04-06
> >
> > Thank you for the author's response. After reading the author's rebuttal, the main concerns I had have been addressed, and I will maintain my score.

---

> > > ### Author Response · Authors · 2025-04-09
> > >
> > > Dear reviewer kFQg,
> > >
> > > We appreciate it a lot that the valuable suggestions and comments you provided. We will incorporate these revisions into our paper.
> > >
> > > Best wishes,
> > > Authors

---

### Official Review · Reviewer_wPXP · 2025-03-14

**Overall Recommendation:** 3

**Summary:**

This study focuses on the semi-supervised learning paradigm within Federated Learning (FL), focused on re-label technology.  Theoretical and empirical results demonstrate the local model has higher relabel accuracy on local data. Furthermore, this paper propose a Bayesian approach to re-label the local data by both the local and global model. Specifically,  bayesian inference to choose model and get weighted combination of pseudo labels. Theoretical analysis shows the lower labeling error and experimental also gives SOTA results.

**Claims And Evidence:**

Yes

**Essential References Not Discussed:**

The related work in ““labels at client” part lacks some recent citations e.g., [R1,R2]

References:

[R1] Zhang, Yonggang, et al. "Robust Training of Federated Models with Extremely Label Deficiency." *The Twelfth International Conference on Learning Representations*.

[R2] Bai, Sikai, et al. "Combating data imbalances in federated semi-supervised learning with dual regulators." *Proceedings of the AAAI conference on artificial intelligence*. Vol. 38. No. 10. 2024.

**Experimental Designs Or Analyses:**

- The experiments about "Impact of Labeled Dataset Size" lack the specific heterogeneity of the experiment, the number of clients, and other specific settings.
- Lacks the $\alpha$ visualization experiments to demonstrate whether the model depends more on the global model to label as the global model improves in performance, so it is recommended to add these experiments.

**Methods And Evaluation Criteria:**

Yes

**Other Comments Or Suggestions:**

See weakness

**Other Strengths And Weaknesses:**

Weakness:
- What’s the meaning of “the global model can progressively improve the re-labeling performance by introducing the extra data knowledge of other clients” in the abstract part? What is the difference between a local model with a high relabel capability and a global model with a progressively higher relabel capability?
- In the motivation part, all experiments do not clearly explain which model to train and what training data and test data are used, so it is a little difficult to follow. For example, how are the global model and local model obtained in Figure 2a, and what are their test data respectively

**Questions For Authors:**

No

**Relation To Broader Scientific Literature:**

New insights about the relationship between the global and local models to relabel data in SSFL.

**Theoretical Claims:**

- It is unclear how we can get the $\hat{p}(x_i,k)$. 5.1 part, the notation are confused, it is better to use a consistent notation to represent this bayesian inference process, because this process is the same for global model and local model.
- In the equation (20), what’s the meaning of $p$ and $\hat{p}$? It is also unclear how to get the equation (20) based on the Hoeffding inequality.

---

> ### Author Rebuttal · Authors · 2025-04-01
>
> Thank you very much for your reviewing and valuable suggestions.
> ```
> Q1: How to get the p^(xi,k) in 5.1. It is better to use a consistent notation.
> ```
> $p^(x_i,k)=p^(x_i,y_i=k)$ represents the joint probability distribution of the sample $x_i$ and $y_i$, i.e., the probability that the sample takes the value $x_i$ while the label simultaneously takes the value $k$. In fact, our notation is consistent throughout the paper. However, due to the relatively large number of variable types involved, it may give an impression of complexity. We will further optimize the explanation and representation of the variable notations moving forward.
>
> ```
> Q2：p and p^ in eq.(20)? It is unclear how to get (20) from Hoeffding.
> ```
> Eq.(20) involves derivation steps that are commonly used in empirical risk analysis, so we omitted these steps. Since $L_p(f)$ is the expectation of $L_{\hat{p}}(f)$, by applying Hoeffding's inequality, we can derive:
> $$
> \mathbb{P}\left( |L_{\hat{p}}(f) - L_p(f)| \geq \epsilon \right) \leq 2 \exp(-2S\epsilon^2)
> $$
> Here, $ \epsilon > 0 $ is an arbitrarily small positive number. Let:
> $$
> \delta = 2 \exp(-2S\epsilon^2)
> $$
> Solving for $ \epsilon $:
> $$
> \epsilon = \sqrt{\frac{\log\frac{2}{\delta}}{2S}}
> $$
> This implies that with a probability of at least $ 1 - \delta $, the following inequality holds:
> $$
> |L_{\hat{p}}(f) - L_p(f)| \leq \sqrt{\frac{\log\frac{2}{\delta}}{2S}}.
> $$
> Decomposing the above absolute value inequality yields formula (20). We will include these in the revised version.
>
> ```
> Q3：Experiments of "Impact of Labeled Dataset Size" lack the specific heterogeneity, the number of clients, and other settings.
> ```
> The total number of clients is 100, with 10% activated in each round. The heterogeneity among clients is set to a shard distribution with \( k=2 \). The threshold is 0.7.
> ```
> Q4:α visualization.
> ```
> We add the following experiment. When each client is selected for training, we record the average value of α for all data points in its local dataset. We here present the corresponding values for five randomly selected clients in the table below. As can be seen, in the initial rounds, the local model achieves higher accuracy on the local data compared to the global model, resulting in a relatively smaller weight α for the global model. In later rounds, the values of α gradually stabilize around 0.5, which aligns with the gradual improvement in the accuracy of the global model.
>
> |Client ID&Round|10|20|40|60|Last|
> |-|-|-|-|-|-|
> |18|0.4531|0.4880|0.5060|0.5146|0.5287|
> |26|0.4643|0.4857|0.4951|0.5101|0.5061|
> |46|0.4477|0.4781|0.4827|0.5045|0.5069|
> |72|0.4585|0.4756|0.4926|0.5088|0.5093|
> |94|0.4602|0.4829|0.4943|0.5023|0.5303|
>
> ```
> Q5:Meaning of “global model can progressively...”. The difference between a local model with a high capability and a global model with a progressively higher capability.
> ```
> - This statement is relative to the local model. In Fig.2b, we observed that when labeling data using only the local model, some labeling errors occur. When these labeled data are used to train the local model, the errors are carried forward, and during the next round of labeling, the same errors persist, making it impossible for the model to self-correct. On the other hand, the global model, which incorporates knowledge from other clients, can correct these errors.
>
> - Since the local model converges quickly while the global model converges more slowly, there is a significant difference in accuracy between the two in the early stages of training, but they gradually converge later on. However, due to the impact of heterogeneity, the error of the global model on the local dataset remains consistently higher than that of the local model.
> ```
> Q6：Motivation does not clearly explain which model to train and what data are used. How are the global and local model obtained in Fig.2a?
> ```
> All motivation experiments used the Wide ResNet28*2 model. As indicated in lines 151 and 208, we used CIFAR10. All experiments were conducted to train the global model.
>
> - In all experiments, the global model is obtained on the server using FedAvg, while the local model is obtained after further local training of the global model.
>
> - In all figures, the global model is tested using the combined local data from all clients (i.e., the global data), whereas the local model is tested using the local data corresponding to its respective client. This is explained in the captions.
>
> Thus:
> - In Fig.2a, as in lines 150–154, the test data for the global model is the global data, while the test data for the local model is the local data corresponding to its respective client.
> - In Fig.2b, the global model is tested using two different annotation methods, and the test data used is the global data.
> - In Fig.2c, the local model is tested using its corresponding local data.
> ```
> Q7: Lacks in some recent citations in “labels at client” [1,2].
> ```
> Thank you for the reminder. We will include them.

---

### Decision · Program_Chairs · 2025-05-01

**Decision:**

Accept (poster)

**Comment:**

After rebuttal, all reviewers agree to accept this paper. To be specific, the proposed method in this paper has been well validated from both theoretical an empirical studies. Besides, the paper is also clearly written.